# Generalization Bounds for Neural Networks via Approximate Description Length

**Amit Daniely**
Hebrew University and Google Research Tel-Aviv
amit.daniely@mail.huji.ac.il

**Elad Granot**
Hebrew University
elad.granot@mail.huji.ac.il

## Abstract

We investigate the sample complexity of networks with bounds on the magnitude of its weights. In particular, we consider the class

$$\mathcal{N} = \{W_t \circ \rho \circ W_{t-1} \circ \rho \ldots \circ \rho \circ W_1 : W_1, \ldots, W_{t-1} \in M_{d \times d}, W_t \in M_{1,d}\}$$

where the spectral norm of each $W_i$ is bounded by $O(1)$, the Frobenius norm is bounded by $R$, and $\rho$ is the sigmoid function $\frac{e^x}{1+e^x}$ or the smoothened ReLU function $\ln(1 + e^x)$. We show that for any depth $t$, if the inputs are in $[-1, 1]^d$, the sample complexity of $\mathcal{N}$ is $\tilde{O}\left(\frac{dR^2}{\epsilon^2}\right)$. This bound is optimal up to log-factors, and substantially improves over the previous state of the art of $\tilde{O}\left(\frac{d^2 R^2}{\epsilon^2}\right)$, that was established in a recent line of work [9, 4, 7, 5, 2, 8].

We furthermore show that this bound remains valid if instead of considering the magnitude of the $W_i$'s, we consider the magnitude of $W_i - W_i^0$, where $W_i^0$ are some reference matrices, with spectral norm of $O(1)$. By taking the $W_i^0$ to be the matrices at the onset of the training process, we get sample complexity bounds that are sub-linear in the number of parameters, in many *typical* regimes of parameters.

To establish our results we develop a new technique to analyze the sample complexity of families $\mathcal{H}$ of predictors. We start by defining a new notion of a randomized approximate description of functions $f : \mathcal{X} \to \mathbb{R}^d$. We then show that if there is a way to approximately describe functions in a class $\mathcal{H}$ using $d$ bits, then $\frac{d}{\epsilon^2}$ examples suffices to guarantee uniform convergence. Namely, that the empirical loss of all the functions in the class is $\epsilon$-close to the true loss. Finally, we develop a set of tools for calculating the approximate description length of classes of functions that can be presented as a composition of linear function classes and non-linear functions.

## 1 Introduction

We analyze the sample complexity of networks with bounds on the magnitude of their weights. Let us consider a prototypical case, where the input space is $\mathcal{X} = [-1, 1]^d$, the output space is $\mathbb{R}$, the number of layers is $t$, all hidden layers has $d$ neurons, and the activation function is $\rho : \mathbb{R} \to \mathbb{R}$. The class of functions computed by such an architecture is

$$\mathcal{N} = \{W_t \circ \rho \circ W_{t-1} \circ \rho \ldots \circ \rho \circ W_1 : W_1, \ldots, W_{t-1} \in M_{d \times d}, W_t \in M_{1,d}\}$$

As the class $\mathcal{N}$ is defined by $(t-1)d^2 + d = O(d^2)$ parameters, classical results (e.g. [1]) tell us that order of $d^2$ examples are sufficient and necessary in order to learn a function from $\mathcal{N}$ (in a standard worst case analysis). However, modern networks often succeed to learn with substantially less examples. One way to provide alternative results, and a potential explanation to the phenomena,

is to take into account the magnitude of the weights. This approach was a success story in the days of SVM [3] and Boosting [10], provided a nice explanation to generalization with sub-linear (in the number of parameters) number of examples, and was even the deriving force behind algorithmic progress. It seems just natural to adopt this approach in the context of modern networks. For instance, it is natural to consider the class

$$\mathcal{N}_R = \{W_t \circ \rho \circ W_{t-1} \circ \rho \ldots \circ \rho \circ W_1 : \forall i, \|W_i\|_F \le R, \|W_i\| \le O(1)\}$$

where $\|W\| = \max_{\|\mathbf{x}\|=1} \|W\mathbf{x}\|$ is the spectral norm and $\|W\|_F = \sqrt{\sum_{i,j=1}^d W_{ij}^2}$ is the Frobenius norm. This class has been analyzed in several recent works [9, 4, 7, 5, 2, 8]. Best known results show a sample complexity of $\tilde{O}\left(\frac{d^2 R^2}{\epsilon^2}\right)$ (for the sake of simplicity, in the introduction, we ignore the dependence on the depth in the big-O notation). In this paper we prove, for various activations, a stronger bound of $\tilde{O}\left(\frac{dR^2}{\epsilon^2}\right)$, which is optimal, up to log factors, for constant depth networks.

How good is this bound? Does it finally provide sub-linear bound in typical regimes of the parameters? To answer this question, we need to ask how large $R$ is. While this question of course don't have a definite answer, empirical studies (e.g. [12]) show that it is usually the case that the norm (spectral, Frobenius, and others) of the weight matrices is at the same order of magnitude as the norm of the matrix in the onset of the training process. In most standard training methods, the initial matrices are random matrices with independent (or almost independent) entries, with mean zero and variance of order $\frac{1}{d}$. The Frobenius norm of such a matrix is of order $\sqrt{d}$. Hence, the magnitude of $R$ is of order $\sqrt{d}$. Going back to our $\tilde{O}\left(\frac{dR^2}{\epsilon^2}\right)$ bound, we get a sample complexity of $\tilde{O}\left(\frac{d^2}{\epsilon^2}\right)$, which is unfortunately still linear in the number of parameters.

Since our bound is almost optimal, we can ask whether this is the end of the story? Should we abandon the aforementioned approach to network sample complexity? A more refined examination of the training process suggests another hope for this approach. Indeed, the training process doesn't start from the zero matrix, but rather form a random initialization matrix. Thus, it stands to reason that instead of considering the magnitude of the weight matrices $W_i$, we should consider the magnitude of $W_i - W_i^0$, where $W_i^0$ is the initial weight matrix. Indeed, empirical studies [6] show that the Frobenius norm of $W_i - W_i^0$ is often order of magnitude smaller than the Frobenius norm of $W_i$. Following this perspective, it is natural to consider the class

$$\mathcal{N}_R(W_1^0, \ldots, W_t^0) = \left\{W_t \circ \rho \circ W_{t-1} \circ \rho \ldots \circ \rho \circ W_1 : \|W_i - W_i^0\| \le O(1), \|W_i - W_i^0\|_F \le R\right\}$$

For some fixed matrices, $W_1^0, \ldots, W_t^0$ of spectral norm[1] $O(1)$. It is natural to expect that considering balls around the initial $W_i^0$'s instead of zero, shouldn't change the sample complexity of the class at hand. In other words, we can expect that the sample complexity of $\mathcal{N}_R(W_1^0, \ldots, W_t^0)$ should be approximately the sample complexity of $\mathcal{N}_R$. Namely, we expect a sample complexity of $\tilde{O}\left(\frac{dR^2}{\epsilon^2}\right)$. Such a bound would finally be sub-linear, as in practice, it is often the case that $R^2 \ll d$.

This approach was pioneered by [4] who considered the class

$$\mathcal{N}_R^{2,1}(W_1^0, \ldots, W_t^0) = \left\{W_t \circ \rho \ldots \circ \rho \circ W_1 : \|W_i - W_i^0\| \le O(1), \|W_i - W_i^0\|_{2,1} \le R\right\}$$

where $\|W\|_{2,1} = \sum_{i=1}^d \sqrt{\sum_{j=1}^d W_{ij}^2}$. For this class they proved a sample complexity bound of $\tilde{O}\left(\frac{dR^2}{\epsilon^2}\right)$. Since, $\|W\|_{2,1} \le \sqrt{d}\|W\|_F$, this implies a sample complexity bound of $\tilde{O}\left(\frac{d^2 R^2}{\epsilon^2}\right)$ on $\mathcal{N}_R(W_1^0, \ldots, W_t^0)$, which is still not sublinear[2]. In this paper we finally prove a sub-linear sample complexity bound of $\tilde{O}\left(\frac{dR^2}{\epsilon^2}\right)$ on $\mathcal{N}_R(W_1^0, \ldots, W_t^0)$.

To prove our results, we develop a new technique for bounding the sample complexity of function classes. Roughly speaking, we define a notion of approximate description of a function, and count

how many bits are required in order to give an approximate description for the functions in the class under study. We then show that this number, called the *approximate description length (ADL)*, gives an upper bound on the sample complexity. The advantage of our method over existing techniques is that it behaves nicely with compositions. That is, once we know the approximate description length of a class $\mathcal{H}$ of functions from $\mathcal{X}$ to $\mathbb{R}^d$, we can also bound the ADL of $\rho \circ \mathcal{H}$, as well as $\mathcal{L} \circ \mathcal{H}$, where $\mathcal{L}$ is a class of linear functions. This allows us to utilize the compositional structure of neural networks.

## 2    Preliminaries

**Notation**    We denote by $\text{med}(x_1, \ldots, x_k)$ the median of $x_1, \ldots, x_k \in \mathbb{R}$. For vectors $\mathbf{x}^1, \ldots, \mathbf{x}^k \in \mathbb{R}^d$ we denote $\text{med}(\mathbf{x}^1, \ldots, \mathbf{x}^k) = \left(\text{med}(x_1^1, \ldots, x_1^k), \ldots, \text{med}(x_d^1, \ldots, x_d^k)\right)$. We use $\log$ to denote $\log_2$, and $\ln$ to denote $\log_e$ An expression of the form $f(n) \lesssim g(n)$ means that there is a universal constant $c > 0$ for which $f(n) \leq cg(n)$. For a finite set $A$ and $f : A \to \mathbb{R}$ we let $\mathbb{E}_{x \in A} f = \mathbb{E}_{x \in A} f(a) = \frac{1}{|A|} \sum_{a \in A} f(a)$. We denote $\mathbb{B}_M^d = \{\mathbf{x} \in \mathbb{R}^d : \|\mathbf{x}\| \leq M\}$ and $\mathbb{B}^d = \mathbb{B}_1^d$. Likewise, we denote $\mathbb{S}^{d-1} = \{\mathbf{x} \in \mathbb{R}^d : \|\mathbf{x}\| = 1\}$ We denote the Frobenius norm of a matrix $W$ by $\|W\|_F^2 = \sum_{ij} W_{ij}^2$, while the spectral norm is denoted by $\|W\| = \max_{\|\mathbf{x}\|=1} \|W\mathbf{x}\|$. For a pair of vectors $\mathbf{x}, \mathbf{y} \in \mathbb{R}^d$ we denote by $\mathbf{xy} \in \mathbb{R}^d$ their point-wise product $\mathbf{xy} = (x_1 y_1, \ldots, x_d y_d)$

**Uniform Convergence and Covering Numbers**    Fix an instance space $\mathcal{X}$, a label space $\mathcal{Y}$ and a loss $\ell : \mathbb{R}^d \times \mathcal{Y} \to [0, \infty)$. We say that $\ell$ is Lipschitz / Bounded / etc. if for any $y \in \mathcal{Y}$, $\ell(\cdot, y)$ is. Fix a class $\mathcal{H}$ from $\mathcal{X}$ to $\mathbb{R}^d$. For a distribution $\mathcal{D}$ and a sample $S \in (\mathcal{X} \times \mathcal{Y})^m$ we define the *representativeness* of $S$ as

$$\text{rep}_\mathcal{D}(S, \mathcal{H}) = \sup_{h \in \mathcal{H}} \ell_\mathcal{D}(h) - \ell_S(h) \text{ for } \ell_\mathcal{D}(h) = \mathbb{E}_{(x,y) \sim \mathcal{D}} \ell(h(x), y) \text{ and } \ell_S(h) = \frac{1}{m} \sum_{i=1}^m \ell(h(x_i), y_i)$$

We note that if $\text{rep}_\mathcal{D}(S, \mathcal{H}) \leq \epsilon$ then any algorithm that is guaranteed to return a function $\hat{h} \in \mathcal{H}$ will enjoy a generalization bound $\ell_\mathcal{D}(h) \leq \ell_S(h) + \epsilon$. In particular, the ERM algorithm will return a function whose loss is optimal, up to an additive factor of $\epsilon$. We will focus on bounds on $\text{rep}_\mathcal{D}(S, \mathcal{H})$ when $S \sim \mathcal{D}^m$. To this end, we will rely on the connection between representativeness and the *covering numbers* of $\mathcal{H}$.

**Definition 2.1.** *Fix a class $\mathcal{H}$ of functions from $\mathcal{X}$ to $\mathbb{R}^d$, an integer $m$, $\epsilon > 0$ and $1 \leq p \leq \infty$. We define $N_p(\mathcal{H}, m, \epsilon)$ as the minimal integer for which the following holds. For every $A \subset \mathcal{X}$ of size $\leq m$ there exists $\tilde{\mathcal{H}} \subset (\mathbb{R}^d)^\mathcal{X}$ such that $\left|\tilde{\mathcal{H}}\right| \leq N_p(\mathcal{H}, m, \epsilon)$ and for any $h \in \mathcal{H}$ there is $\tilde{h} \in \tilde{\mathcal{H}}$ with*
$$\left(\mathbb{E}_{x \in A} \left\|h(x) - \tilde{h}(x)\right\|_\infty^p\right)^{\frac{1}{p}} \leq \epsilon. \text{ For } p = 2, \text{ we denote } N(\mathcal{H}, m, \epsilon) = N_2(\mathcal{H}, m, \epsilon)$$

We conclude with a lemma, which will be useful in this paper. The proof can be found in the supplementary material.

**Lemma 2.2.** *Let $\ell : \mathbb{R}^d \times \mathcal{Y} \to \mathbb{R}$ be L-Lipschitz w.r.t. $\|\cdot\|_\infty$ and B-bounded. Assume that for any $0 < \epsilon \leq 1$, $\log\left(N(\mathcal{H}, m, \epsilon)\right) \leq \frac{n}{\epsilon^2}$. Then $\mathbb{E}_{S \sim \mathcal{D}^m} \text{rep}_\mathcal{D}(S, \mathcal{H}) \lesssim \frac{(L+B)\sqrt{n}}{\sqrt{m}} \log(m)$. Furthermore, with probability at least $1 - \delta$, $\text{rep}_\mathcal{D}(S, \mathcal{H}) \lesssim \frac{(L+B)\sqrt{n}}{\sqrt{m}} \log(m) + B\sqrt{\frac{2 \ln(2/\delta)}{m}}$*

**A Basic Inequality**

**Lemma 2.3.** *Let $X_1, \ldots, X_n$ be independent r.v. with that that are $\sigma$-estimators to $\mu$. Then $\Pr\left(|\text{med}(X_1, \ldots, X_n) - \mu| > k\sigma\right) < \left(\frac{2}{k}\right)^n$*

## 3    Simplified Approximate Description Length

To give a soft introduction to our techniques, we first consider a simplified version of it. We next define the *approximate description length* of a class $\mathcal{H}$ of functions from $\mathcal{X}$ to $\mathbb{R}^d$, which quantifies the number of bits it takes to approximately describe a function from $\mathcal{H}$. We will use the following notion of approximation

**Definition 3.1.** *A random vector $X \in \mathbb{R}^d$ is a $\sigma$-estimator to $\mathbf{x} \in \mathbb{R}^d$ if*

$$\mathbb{E}\, X = \mathbf{x} \text{ and } \forall \mathbf{u} \in \mathbb{S}^{d-1},\ \mathrm{VAR}(\langle \mathbf{u}, X \rangle) = \mathbb{E}\, \langle \mathbf{u}, X - \mathbf{x} \rangle^2 \le \sigma^2$$

*A random function $\hat{f} : \mathcal{X} \to \mathbb{R}^d$ is a $\sigma$-estimator to $f : \mathcal{X} \to \mathbb{R}^d$ if for any $x \in \mathcal{X}$, $\hat{f}(x)$ is a $\sigma$-estimator to $f(x)$.*

A $(\sigma, n)$-*compressor* $\mathcal{C}$ for a class $\mathcal{H}$ takes as input a function $h \in \mathcal{H}$, and outputs a (random) function $\mathcal{C}h$ such that (i) $\mathcal{C}h$ is a $\sigma$-estimator of $h$ and (ii) it takes $n$ bits to describe $\mathcal{C}h$. Formally,

**Definition 3.2.** *A $(\sigma, n)$-compressor for $\mathcal{H}$ is a triplet $(\mathcal{C}, \Omega, \mu)$ where $\mu$ is a probability measure on $\Omega$, and $\mathcal{C}$ is a function $\mathcal{C} : \Omega \times \mathcal{H} \to \left(\mathbb{R}^d\right)^{\mathcal{X}}$ such that*

1. *For any $h \in \mathcal{H}$ and $x \in \mathcal{X}$, $(\mathcal{C}_\omega h)(x)$, $\omega \sim \mu$ is a $\sigma$-estimator of $h(x)$.*

2. *There are functions $E : \Omega \times \mathcal{H} \to \{\pm 1\}^n$ and $D : \{\pm 1\}^n \to \left(\mathbb{R}^d\right)^{\mathcal{X}}$ for which $\mathcal{C} = D \circ E$*

**Definition 3.3.** *We say that a class $\mathcal{H}$ of functions from $\mathcal{X}$ to $\mathbb{R}^d$ has* approximate description length *$n$ if there exists a $(1, n)$-compressor for $\mathcal{H}$*

It is not hard to see that if $(\mathcal{C}, \Omega, \mu)$ is a $(\sigma, n)$-compressor for $\mathcal{H}$, then

$$(\mathcal{C}_{\omega_1, \ldots, \omega_k} h)(x) := \frac{\sum_{i=1}^{k} (\mathcal{C}_{\omega_i} h)(x)}{k}$$

is a $\left(\frac{\sigma}{\sqrt{k}}, kn\right)$-compressor for $\mathcal{H}$. Hence, if the approximate description length of $\mathcal{H}$ is $n$, then for any $1 \ge \epsilon > 0$ there exists an $\left(\epsilon, n\lceil \epsilon^{-2} \rceil\right)$-compressor for $\mathcal{H}$.

We next connect the approximate description length, to covering numbers and representativeness. We separate it into two lemmas, one for $d = 1$ and one for general $d$, as for $d = 1$ we can prove a slightly stronger bound.

**Lemma 3.4.** *Fix a class $\mathcal{H}$ of functions from $\mathcal{X}$ to $\mathbb{R}$ with approximate description length $n$. Then, $\log\left(N(\mathcal{H}, m, \epsilon)\right) \le n \left\lceil \epsilon^{-2} \right\rceil$. Hence, if $\ell : \mathbb{R}^d \times \mathcal{Y} \to \mathbb{R}$ is $L$-Lipschitz and $B$-bounded, then for any distribution $\mathcal{D}$ on $\mathcal{X} \times \mathcal{Y}$, $\mathbb{E}_{S \sim \mathcal{D}^m} \mathrm{rep}_\mathcal{D}(S, \mathcal{H}) \lesssim \frac{(L+B)\sqrt{n}}{\sqrt{m}} \log(m)$. Furthermore, with probability at least $1 - \delta$, $\mathrm{rep}_\mathcal{D}(S, \mathcal{H}) \lesssim \frac{(L+B)\sqrt{n}}{\sqrt{m}} \log(m) + B\sqrt{\frac{2\ln(2/\delta)}{m}}$*

**Lemma 3.5.** *Fix a class $\mathcal{H}$ of functions from $\mathcal{X}$ to $\mathbb{R}^d$ with approximate description length $n$. Then,*
$$\log\left(N_\infty(\mathcal{H}, m, \epsilon)\right) \le \log\left(N(\mathcal{H}, m, \epsilon)\right) \le n \left\lceil 16\epsilon^{-2} \right\rceil \left\lceil \log(dm) \right\rceil$$

*Hence, if $\ell : \mathbb{R}^d \times \mathcal{Y} \to \mathbb{R}$ is $L$-Lipschitz w.r.t. $\| \cdot \|_\infty$ and $B$-bounded, then for any distribution $\mathcal{D}$ on $\mathcal{X} \times \mathcal{Y}$, $\mathbb{E}_{S \sim \mathcal{D}^m} \mathrm{rep}_\mathcal{D}(S, \mathcal{H}) \lesssim \frac{(L+B)\sqrt{n \log(dm)}}{\sqrt{m}} \log(m)$. Furthermore, with probability at least $1 - \delta$, $\mathrm{rep}_\mathcal{D}(S, \mathcal{H}) \lesssim \frac{(L+B)\sqrt{n \log(dm)}}{\sqrt{m}} \log(m) + B\sqrt{\frac{2\ln(2/\delta)}{m}}$*

## 3.1 Linear Functions

We next bound the approximate description length of linear functions with bounded Frobenius norm.

**Theorem 3.6.** *Let class $\mathcal{L}_{d_1, d_2, M} = \left\{ \mathbf{x} \in \mathbb{B}^{d_1} \mapsto W\mathbf{x} : W \text{ is } d_2 \times d_1 \text{ matrix with } \|W\|_F \le M \right\}$ has approximate description length*

$$n \le \left\lceil \frac{1}{4} + 2M^2 \right\rceil 2 \left\lceil \log\left(2d_1 d_2 (M+1)\right) \right\rceil$$

*Hence, if $\ell : \mathbb{R}^{d_2} \times \mathcal{Y} \to \mathbb{R}$ is $L$-Lipschitz w.r.t. $\| \cdot \|_\infty$ and $B$-bounded, then for any distribution $\mathcal{D}$ on $\mathcal{X} \times \mathcal{Y}$*

$$\mathbb{E}_{S \sim \mathcal{D}^m} \mathrm{rep}_\mathcal{D}(S, \mathcal{L}_{d_1, d_2, M}) \lesssim \frac{(L+B)\sqrt{M^2 \log(d_1 d_2 M) \log(d_2 m)}}{\sqrt{m}} \log(m)$$

*Furthermore, with probability at least $1 - \delta$,*

$$\mathrm{rep}_\mathcal{D}(S, \mathcal{L}_{d_1, d_2, M}) \lesssim \frac{(L+B)\sqrt{M^2 \log(d_1 d_2 M) \log(d_2 m)}}{\sqrt{m}} \log(m) + B\sqrt{\frac{2\ln(2/\delta)}{m}}$$

We remark that the above bounds on the representativeness coincides with standard bounds ([11] for instance), up to log factors. The advantage of these bound is that they remain valid for *any output dimension $d_2$*.

In order to prove theorem 3.6 we will use a randomized sketch of a matrix.

**Definition 3.7.** *Let* $\mathbf{w} \in \mathbb{R}^d$ *be a vector. A* random sketch of $\mathbf{w}$ *is a random vector $\hat{\mathbf{w}}$ that is samples as follows. Choose $i$ w.p. $p_i = \frac{w_i^2}{2\|\mathbf{w}\|^2} + \frac{1}{2d}$. Then, w.p. $\frac{w_i}{p_i} - \left\lfloor \frac{w_i}{p_i} \right\rfloor$ let $b = 1$ and otherwise $b = 0$. Finally, let $\hat{\mathbf{w}} = \left( \left\lfloor \frac{w_i}{p_i} \right\rfloor + b \right) \mathbf{e}_i$. A* random $k$-sketch of $\mathbf{w}$ *is an average of $k$-independent random sketches of $\mathbf{w}$. A random sketch and a random $k$-sketch of a matrix is defined similarly, with the standard matrix basis instead of the standard vector basis.*

The following useful lemma shows that an sketch $\mathbf{w}$ is a $\sqrt{\frac{1}{4} + 2\|\mathbf{w}\|^2}$-estimator of $\mathbf{w}$.

**Lemma 3.8.** *Let $\hat{\mathbf{w}}$ be a random sketch of $\mathbf{w} \in \mathbb{R}^d$. Then, (1) $\mathbb{E}\, \hat{\mathbf{w}} = \mathbf{w}$ and (2) for any $\mathbf{u} \in \mathbb{S}^{d-1}$,*
$\mathbb{E}\left(\langle \mathbf{u}, \hat{\mathbf{w}} \rangle - \langle \mathbf{u}, \mathbf{w} \rangle\right)^2 \leq \mathbb{E}\, \langle \mathbf{u}, \hat{\mathbf{w}} \rangle^2 \leq \frac{1}{4} + 2\|\mathbf{w}\|^2$

*Proof.* (of theorem 3.6) We construct a compressor for $\mathcal{L}_{d_1,d_2,M}$ as follows. Given $W$, we will sample a $k$-sketch $\hat{W}$ of $W$ for $k = \left\lceil \frac{1}{4} + 2M^2 \right\rceil$, and will return the function $\mathbf{x} \mapsto \hat{W}\mathbf{x}$. We claim that that $W \mapsto \hat{W}$ is a $(1, 2k \lceil \log(2d_1 d_2(M+1)) \rceil)$-compressor for $\mathcal{L}_{d_1,d_2,M}$. Indeed, to specify a sketch of $W$ we need $\lceil \log(d_1 d_2) \rceil$ bits to describe the chosen index, as well as $\log(2d_1 d_2 M + 2)$ bits to describe the value in that index. Hence, $2k \lceil \log(2d_1 d_2(M+1)) \rceil$ bits suffices to specify a $k$-sketch. It remains to show that for $\mathbf{x} \in \mathbb{B}^{d_1}$, $\hat{W}\mathbf{x}$ is a 1-estimator of $W\mathbf{x}$. Indeed, by lemma 3.8, $\mathbb{E}\, \hat{W} = W$ and therefore $\mathbb{E}\, \hat{W}\mathbf{x} = W\mathbf{x}$. Likewise, for $\mathbf{u} \in \mathbb{S}^{d_2-1}$. We have

$$\mathbb{E}\left(\left\langle \mathbf{u}, \hat{W}\mathbf{x} \right\rangle - \langle \mathbf{u}, W\mathbf{x} \rangle\right)^2 = \mathbb{E}\left(\left\langle \hat{W}, \mathbf{x}\mathbf{u}^T \right\rangle - \left\langle W, \mathbf{x}\mathbf{u}^T \right\rangle\right)^2 \leq \frac{\frac{1}{4} + 2M^2}{k} \leq 1$$

$\square$

## 3.2 Simplified Depth 2 Networks

To demonstrate our techniques, we consider the following class of functions. We let the domain $\mathcal{X}$ to be $\mathbb{B}^d$. We fix an activation function $\rho : \mathbb{R} \to \mathbb{R}$ that is assumed to be a polynomial $\rho(x) = \sum_{i=0}^{k} a_i x^i$ with $\sum_{n=1}^{n} |a_n| = 1$. For any $W \in M_{d,d}$ we define $h_W(\mathbf{x}) = \frac{1}{\sqrt{d}} \sum_{i=1}^{d} \rho(\langle \mathbf{w}_i, \mathbf{x} \rangle)$ Finally, we let $\mathcal{H} = \left\{ h_W : \forall i, \|\mathbf{w}_i\| \leq \frac{1}{2} \right\}$ In order to build compressors for classes of networks, we will utilize to compositional structure of the classes. Specifically, we have that $\mathcal{H} = \Lambda \circ \rho \circ \mathcal{F}$ where $\mathcal{F} = \{x \mapsto W\mathbf{x} : W \text{ is } d \times d \text{ matrix with } \|\mathbf{w}_i\| \leq 1 \text{ for all } i\}$ and $\Lambda(\mathbf{x}) = \frac{1}{\sqrt{d}} \sum_{i=1}^{d} x_i$.

As $\mathcal{F}$ is a subset of $\mathcal{L}_{d,d,\sqrt{d}}$, we know that there exists a $(1, O\left(d \log(d)\right))$-compressor for it. We will use this compressor to build a compressor to $\rho \circ \mathcal{F}$, and then to $\Lambda \circ \rho \circ \mathcal{F}$. We will start with the latter, linear case, which is simpler

**Lemma 3.9.** *Let $X$ be a $\sigma$-estimator to $\mathbf{x} \in \mathbb{R}^{d_1}$. Let $A \in M_{d_2,d_1}$ be a matrix of spectral norm $\leq r$. Then, $AX$ is a $(r\sigma)$-estimator to $A\mathbf{x}$. In particular, if $\mathcal{C}$ is a $(1, n)$-compressor to a class $\mathcal{H}$ of functions from $\mathcal{X}$ to $\mathbb{R}^d$. Then*
$$\mathcal{C}'_\omega(\Lambda \circ h) = \Lambda \circ \mathcal{C}_\omega h$$
*is a $(1, n)$-compressor to $\Lambda \circ \mathcal{H}$*

We next consider the composition of $\mathcal{F}$ with the non-linear $\rho$. As opposed to composition with a linear function, we cannot just generate a compression version using $\mathcal{F}$'s compressor and then compose with $\rho$. Indeed, if $X$ is a $\sigma$-estimator to $\mathbf{x}$, it is not true in general that $\rho(X)$ is an estimator of $\rho(\mathbf{x})$. For instance, consider the case that $\rho(x) = x^2$, and $X = (X_1, \ldots, X_d)$ is a vector of independent standard Gaussians. $X$ is a 1-estimator of $0 \in \mathbb{R}^d$. On the other hand, $\rho(X) = (X_1^2, \ldots, X_n^2)$ is not an estimator of $0 = \rho(0)$. We will therefore take a different approach. Given $f \in \mathcal{F}$, we will sample $k$ independent estimators $\{C_{\omega_i} f\}_{i=1}^k$ from $\mathcal{F}$'s compressor, and define the compressed version of $\sigma \circ h$ as $\mathcal{C}'_{\omega_1,\ldots,\omega_k} f = \sum_{i=0}^{d} a_i \prod_{j=0}^{i} C_{\omega_i} f$. This construction is analyzed in the following lemma

**Lemma 3.10.** *If $\mathcal{C}$ is a $\left(\frac{1}{2}, n\right)$-compressor of a class $\mathcal{H}$ of functions from $\mathcal{X}$ to $\left[-\frac{1}{2}, \frac{1}{2}\right]^d$. Then $\mathcal{C}'$ is a $(1, n)$-compressor of $\rho \circ \mathcal{H}$*

Combining theorem 3.6 and lemmas 3.9, 3.10 we have:

**Theorem 3.11.** *$\mathcal{H}$ has approximation length $\lesssim d \log(d)$. Hence, if $\ell : \mathbb{R} \times \mathcal{Y} \to \mathbb{R}$ is L-Lipschitz and B-bounded, then for any distribution $\mathcal{D}$ on $\mathcal{X} \times \mathcal{Y}$*

$$\mathop{\mathbb{E}}_{S \sim \mathcal{D}^m} \operatorname{rep}_{\mathcal{D}}(S, \mathcal{H}) \lesssim \frac{(L+B)\sqrt{d \log(d)}}{\sqrt{m}} \log(m)$$

*Furthermore, with probability at least $1 - \delta$,*

$$\operatorname{rep}_{\mathcal{D}}(S, \mathcal{H}) \lesssim \frac{(L+B)\sqrt{d \log(d)}}{\sqrt{m}} \log(m) + B\sqrt{\frac{2 \ln(2/\delta)}{m}}$$

Lemma 3.10 is implied by the following useful lemma:

**Lemma 3.12.**     *1. If $X$ is a $\sigma$-estimator of $\mathbf{x}$ then $aX$ is a $(|a|\sigma)$-estimator of $aX$*

2. *Suppose that for $n = 1, 2, 3, \ldots$ $X_n$ is a $\sigma_n$-estimator of $\mathbf{x}_n \in \mathbb{R}^d$. Assume furthermore that $\sum_{n=1}^{\infty} \mathbf{x}_n$ and $\sum_{n=1}^{\infty} \sigma_n$ converge to $\mathbf{x} \in \mathbb{R}^d$ and $\sigma \in [0, \infty)$. Then, $\sum_{n=1}^{\infty} X_n$ is a $\sigma$-estimator of $\mathbf{x}$*

3. *Suppose that $\{X_i\}_{i=1}^{k}$ are independent $\sigma_i$-estimators of $\mathbf{x}_i \in \mathbb{R}^d$. Then $\prod_{i=1}^{k} X_i$ is a $\sigma'$-estimator of $\prod_{i=1}^{k} \mathbf{x}_i$ for $\sigma'^2 = \prod_{i=1}^{k} \left(\sigma_i^2 + \|\mathbf{x}_i\|_{\infty}^2\right) - \prod_{i=1}^{k} \|\mathbf{x}_i\|_{\infty}^2$*

We note that the bounds in the above lemma are all tight.

# 4   Approximation Description Length

In this section we refine the definition of approximate description length that were given in section 3. We start with the encoding of the compressed version of the functions. Instead of standard strings, we will use what we call *bracketed string*. The reason for that often, in order to create a compressed version of a function, we concatenate compressed versions of other functions. This results with strings with a nested structure. For instance, consider the case that a function $h$ is encoded by the concatenation of $h_1$ and $h_2$. Furthermore, assume that $h_1$ is encoded by the string $01$, while $h_2$ is encoded by the concatenation of $h_3, h_4$ and $h_5$ that are in turn encoded by the strings $101$, $0101$ and $1110$. The encoding of $h$ will then be $[[01][[101][0101][1110]]]$. We note that in section 3 we could avoid this issue since the length of the strings and the recursive structure were fixed, and did not depend on the function we try to compress. Formally, we define

**Definition 4.1.** *A* bracketed string *is a rooted tree $S$, such that (i) the children of each edge are ordered, (ii) there are no nodes with a singe child, and (iii) the leaves are labeled by $\{0, 1\}$. The length, $\operatorname{len}(S)$ of $S$ is the number of its leaves.*

Let $S$ be a bracketed string. There is a linear order on its leaves that is defined as follows. Fix a pair of leaves, $v_1$ and $v_2$, and let $u$ be their LCA. Let $u_1$ (resp. $u_2$) be the child of $u$ that lie on the path to $v_1$ (resp. $v_2$). We define $v_1 < v_2$ if $u_1 < u_2$ and $v_1 > v_2$ otherwise (note that necessarily $u_1 \neq u_2$). Let $v_1, \ldots, v_n$ be the leaves of $T$, ordered according to the above order, and let $b_1, \ldots, b_n$ be the corresponding bits. The string associated with $T$ is $s = b_1 \ldots b_n$. We denote by $\mathcal{S}_n$ the collection of bracketed strings of length $\leq n$, and by $\mathcal{S} = \cup_{n=1}^{\infty} \mathcal{S}_n$ the collection of all bracketed strings.

The following lemma shows that in log-scale, the number of bracketed strings of length $\leq n$ differ from standard strings of length $\leq n$ by only a constant factor

**Lemma 4.2.** $|\mathcal{S}_n| \leq 32^n$

We next revisit the definition of a compressor for a class $\mathcal{H}$. The definition of compressor will now have a third parameter, $n_s$, in addition to $\sigma$ and $n$. We will make three changes in the definition. The first, which is only for the sake of convenience, is that we will use bracketed strings rather than standard strings. The second change, is that the length of the encoding string will be bounded only

*in expectation.* The final change is that the compressor can now output a *seed*. That is, given a function $h \in \mathcal{H}$ that we want to compress, the compressor can generate both a non-random seed $E_s(h) \in \mathcal{S}_{n_s}$ and a random encoding $E(\omega, h) \in \mathcal{S}$ with $\mathbb{E}_{\omega \sim \mu} \operatorname{len}(E(\omega, h)) \leq n$. Together, $E_s(h)$ and $E(\omega, h)$ encode a $\sigma$-estimator. Namely, there is a function $D : \mathcal{S}_{n_s} \times \mathcal{S} \to (\mathbb{R}^d)^{\mathcal{X}}$ such that $D(E_s(h), E(\omega, h))$, $\omega \sim \mu$ is a $\sigma$-estimator of $h$. The advantage of using seeds is that it will allow us to generate many independent estimators, at a lower cost. In the case that $n \ll n_s$, the cost of generating $k$ independent estimators of $h \in \mathcal{H}$ is $n_s + kn$ bits (in expectation) instead of $k(n_s + n)$ bits. Indeed, we can encode $k$ estimators by a single seed $E_s(h)$ and $k$ independent "regular" encodings $E(\omega_k, h), \dots, E(\omega_k, h)$. The formal definition is given next.

**Definition 4.3.** *A $(\sigma, n_s, n)$-compressor for $\mathcal{H}$ is a 5-tuple $\mathcal{C} = (E_s, E, D, \Omega, \mu)$ where $\mu$ is a probability measure on $\Omega$, and $E_s, E, D$ are functions $E_s : \mathcal{H} \to \mathcal{T}^{n_s}$, $E : \Omega \times \mathcal{H} \to \mathcal{T}$, and $D : \mathcal{T}^{n_s} \times \mathcal{T} \to (\mathbb{R}^d)^{\mathcal{X}}$ such that for any $h \in \mathcal{H}$ and $x \in \mathcal{X}$ (1) $D(E_s(h), E(\omega, h))$, $\omega \sim \mu$ is a $\sigma$-estimator of $h$ and (2) $\mathbb{E}_{\omega \sim \mu} \operatorname{len}(E(\omega, h)) \leq n$*

We finally revisit the definition of approximate description length. We will add an additional parameter, to accommodate the use of seeds. Likewise, the approximate description length will now be a function of $m$ – we will say that $\mathcal{H}$ has approximate description length $(n_s(m), n(m))$ if there is a $(1, n_s(m), n(m))$-compressor for the restriction of $\mathcal{H}$ to any set $A \subset \mathcal{X}$ of size at most $m$. Formally:

**Definition 4.4.** *We say that a class $\mathcal{H}$ of functions from $\mathcal{X}$ to $\mathbb{R}^d$ has* approximate description length *$(n_s(m), n(m))$ if for any set $A \subset \mathcal{X}$ of size $\leq m$ there exists a $(1, n_s(m), n(m))$-compressor for $\mathcal{H}|_A$*

It is not hard to see that if $\mathcal{H}$ has approximate description length $(n_s(m), n(m))$, then for any $1 \geq \epsilon > 0$ and a set $A \subset \mathcal{X}$ of size $\leq m$, there exists an $\left(\epsilon, n_s(m), n(m)\lceil \epsilon^{-2} \rceil\right)$-compressor for $\mathcal{H}|_A$. We next connect the approximate description length, to covering numbers and representativeness. The proofs are similar the the proofs of lemmas 3.4 and 3.5.

**Lemma 4.5.** *Fix a class $\mathcal{H}$ of functions from $\mathcal{X}$ to $\mathbb{R}$ with approximate description length $(n_s(m), n(m))$. Then, $\log(N(\mathcal{H}, m, \epsilon)) \lesssim n_s(m) + \frac{n(m)}{\epsilon^2}$ Hence, if $\ell : \mathbb{R}^d \times \mathcal{Y} \to \mathbb{R}$ is $L$-Lipschitz and $B$-bounded, then for any distribution $\mathcal{D}$ on $\mathcal{X} \times \mathcal{Y}$*

$$\mathbb{E}_{S \sim \mathcal{D}^m} \operatorname{rep}_\mathcal{D}(S, \mathcal{H}) \lesssim \frac{(L + B)\sqrt{n_s(m) + n(m)}}{\sqrt{m}} \log(m)$$

*Furthermore, with probability at least $1 - \delta$,*

$$\operatorname{rep}_\mathcal{D}(S, \mathcal{H}) \lesssim \frac{(L + B)\sqrt{n_s(m) + n(m)}}{\sqrt{m}} \log(m) + B\sqrt{\frac{2 \ln(2/\delta)}{m}}$$

**Lemma 4.6.** *Fix a class $\mathcal{H}$ of functions from $\mathcal{X}$ to $\mathbb{R}^d$ with approximate description length $(n_s(m), n(m))$. Then, $\log(N(\mathcal{H}, m, \epsilon)) \leq \log(N_\infty(\mathcal{H}, m, \epsilon)) \lesssim n_s(m) + \frac{n(m) \log(dm)}{\epsilon^2}$. Hence, if $\ell : \mathbb{R}^d \times \mathcal{Y} \to \mathbb{R}$ is $L$-Lipschitz w.r.t. $\| \cdot \|_\infty$ and $B$-bounded, then for any distribution $\mathcal{D}$ on $\mathcal{X} \times \mathcal{Y}$*

$$\mathbb{E}_{S \sim \mathcal{D}^m} \operatorname{rep}_\mathcal{D}(S, \mathcal{H}) \lesssim \frac{(L + B)\sqrt{n_s(m) + n(m) \log(dm)}}{\sqrt{m}} \log(m)$$

*Furthermore, with probability at least $1 - \delta$,*

$$\operatorname{rep}_\mathcal{D}(S, \mathcal{H}) \lesssim \frac{(L + B)\sqrt{n_s(m) + n(m) \log(dm)}}{\sqrt{m}} \log(m) + B\sqrt{\frac{2 \ln(2/\delta)}{m}}$$

We next analyze the behavior of the approximate description length under various operations

**Lemma 4.7.** *Let $\mathcal{H}_1, \mathcal{H}_2$ be classes of functions from $\mathcal{X}$ to $\mathbb{R}^d$ with approximate description length of $(n_s^1(m), n^1(m))$ and $(n_s^2(m), n^2(m))$. Then $\mathcal{H}_1 + \mathcal{H}_2$ has approximate description length of $(n_s^1(m) + n_s^2(m), 2n^1(m) + 2n^2(m))$*

**Lemma 4.8.** *Let $\mathcal{H}$ be a class of functions from $\mathcal{X}$ to $\mathbb{R}^d$ with approximate description length of $(n_s(m), n(m))$. Let $A$ be $d_2 \times d_1$ matrix. Then $A \circ \mathcal{H}_1$ has approximate description length $\left(n_s(m), \lceil \|A\|^2 \rceil n(m)\right)$*

Figure 1: The functions $\ln(1 + e^x)$ and $\frac{e^x}{1+e^x}$

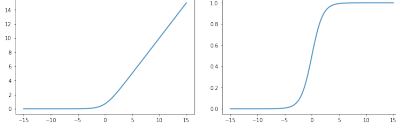

**Definition 4.9.** *Denote by $\mathcal{L}_{d_1,d_2,r,R}$ the class of all $d_2 \times d_1$ matrices of spectral norm at most $r$ and Frobenius norm at most $R$.*

**Lemma 4.10.** *Let $\mathcal{H}$ be a class of functions from $\mathcal{X}$ to $\mathbb{R}^{d_1}$ with approximate description length $(n_s(m), n(m))$. Assume furthermore that for any $x \in \mathcal{X}$ and $h \in \mathcal{H}$ we have that $\|h(x)\| \le B$. Then, $\mathcal{L}_{d_1,d_2,r,R} \circ \mathcal{H}$ has approximate description length*

$$\left(n_s(m), n(m)O(r^2 + 1) + O\left((d_1 + B^2)(R^2 + 1)\log(Rd_1d_2 + 1)\right)\right)$$

**Definition 4.11.** *A function $f : \mathbb{R} \to \mathbb{R}$ is $B$-strongly-bounded if for all $n \ge 1$, $\|f^{(n)}\|_\infty \le n!B^n$. Likewise, $f$ is strongly-bounded if it is $B$-strongly-bounded for some $B$*

We note that

**Lemma 4.12.** *If $f$ is $B$-strongly-bounded then $f$ is analytic and its Taylor coefficients around any point are bounded by $B^n$*

The following lemma gives an example to a strongly bounded sigmoid function, as well as a strongly bounded smoothened version of the ReLU (see figure 1).

**Lemma 4.13.** *The functions $\ln(1 + e^x)$ and $\frac{e^x}{1+e^x}$ are strongly-bounded*

**Lemma 4.14.** *Let $\mathcal{H}$ be a class of functions from $\mathcal{X}$ to $\mathbb{R}^d$ with approximate description length of $(n_s(m), n(m))$. Let $\rho : \mathbb{R} \to \mathbb{R}$ be $B$-strongly-bounded. Then, $\rho \circ \mathcal{H}$ has approximate description length of*

$$\left(n_s(m) + O\left(n(m)B^2\log(md)\right), O\left(n(m)B^2\log(d)\right)\right)$$

# 5 Sample Complexity of Neural Networks

Fix the instance space $\mathcal{X}$ to be the ball of radius $\sqrt{d}$ in $\mathbb{R}^d$ (in particular $[-1,1]^d \subset \mathcal{X}$) and a $B$-strongly-bounded activation $\rho$. Fix matrices $W_i^0 \in M_{d_i,d_{i-1}}$, $i = 1, \ldots, t$. Consider the following class of depth-$t$ networks

$$\mathcal{N}_{r,R}(W_1^0, \ldots, W_t^0) = \left\{W_t \circ \rho \circ W_{t-1} \circ \rho \ldots \circ \rho \circ W_1 : \|W_i - W_i^0\| \le r, \|W_i - W_i^0\|_F \le R\right\}$$

We note that

$$\mathcal{N}_{r,R}(W_1^0, \ldots, W_t^0) = \mathcal{N}_{r,R}(W_t^0) \circ \ldots \circ \mathcal{N}_{r,R}(W_1^0)$$

The following lemma analyzes the cost, in terms of approximate description length, when moving from a class $\mathcal{H}$ to $\mathcal{N}_{r,R}(W^0) \circ \mathcal{H}$.

**Lemma 5.1.** *Let $\mathcal{H}$ be a class of functions from $\mathcal{X}$ to $\mathbb{R}^{d_1}$ with approximate description length $(n_s(m), n(m))$ and $\|h(x)\| \le M$ for any $x \in \mathcal{X}$ and $h \in \mathcal{H}$. Fix $W^0 \in M_{d_2,d_1}$. Then, $\mathcal{N}_{r,R}(W_t^0) \circ \mathcal{H}$ has approximate description length of*

$$\left(n_s(m) + n'(m)B^2\log(md_2), n'(m)B^2\log(d_2)\right)$$

*for*

$$n'(m) = n(m)O(r^2 + \|W^0\|^2 + 1) + O\left((d_1 + M^2)(R^2 + 1)\log(Rd_1d_2 + 1)\right)$$

The lemma is follows by combining lemmas 4.7, 4.8, 4.10 and 4.14. We note that in the case that $d_1, d_2 \le d$, $M = O(\sqrt{d_1})$, $B, r, \|W^0\| = O(1)$ (and hence $R = O\left(\sqrt{d}\right)$) and $R \ge 1$ we get that $\mathcal{N}_{r,R}(W^0) \circ \mathcal{H}$ has approximate description length of

$$\left(n_s(m) + O\left(n(m)\log(md)\right), O\left(n(m)\log(d)\right) + O\left(d_1R^2\log^2(d)\right)\right)$$

By induction, the approximate description length of $\mathcal{N}_{r,R}(W_1^0, \ldots, W_t^0)$ is

$$\left(dR^2O\left(\log(d)\right)^t\log(md), dR^2O\left(\log(d)\right)^{t+1}\right)$$

## Footnotes

[1]The bound of $O(1)$ on the spectral norm of the $W_i^0$'s and $W_i - W_i^0$ is again motivated by the practice of neural networks – the spectral norm of $W_i^0$, with standard initializations, is $O(1)$, and empirical studies [6, 12] show that the spectral norm of $W_i - W_i^0$ is usually very small.

[2]We note that $\|W\|_{2,1} = \Theta(\sqrt{d})$ even if $W$ is a random matrix with variance that is calibrated so that $\|W\|_F = \Theta(1)$ (namely, each entry has variance $\frac{1}{d^2}$).

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
