[Supplementary Material]

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

}) \le B2^{-M+1} + \frac{12B}{\sqrt{m}} \sum_{k=1}^{M} 2^{-k}\sqrt{\ln\left(N(\ell\circ\mathcal{H}, m, B2^{-k})\right)}$$

*Furthermore, with probability at least $1-\delta$,*

$$\mathrm{rep}_{\mathcal{D}}(S,\mathcal{H}) \le B2^{-M+1} + \frac{12B}{\sqrt{m}} \sum_{k=1}^{M} 2^{-k}\sqrt{\ln\left(N(\ell\circ\mathcal{H}, m, B2^{-k})\right)} + B\sqrt{\frac{2\ln(2/\delta)}{m}}$$

*Proof.* (of lemma 2.2) Denote

$$A = B2^{-M+1} + \frac{12B}{\sqrt{m}} \sum_{k=1}^{M} 2^{-k}\sqrt{\ln\left(N(\ell\circ\mathcal{H}, m, B2^{-k})\right)}$$

We will show that $A \lesssim \frac{(L+B)\sqrt{n}}{\sqrt{m}}\log(m)$. We have, if $\frac{B2^{-k}}{L} \le 1$,

$$\ln\left(N(\ell\circ\mathcal{H}, m, B2^{-k})\right) \le \ln\left(N\left(\mathcal{H}, m, \frac{B}{L}2^{-k}\right)\right) \le \frac{nL^2 2^{2k}}{B^2} + n$$

Hence,

$$A \le B2^{-M+1} + \frac{12B}{\sqrt{m}}\sum_{k=1}^{M}\frac{\sqrt{n}L}{B} + \frac{12B}{\sqrt{m}}\sum_{k=1}^{M}2^{-k}\sqrt{n} \le B2^{-M+1} + \frac{12(LM+B)\sqrt{n}}{\sqrt{m}}$$

Choosing $M = \log\left(\sqrt{\frac{m}{n}}\right)$ we get,

$$A \le \frac{12(L\log\left(\sqrt{\frac{m}{n}}\right)+B)\sqrt{n} + B\sqrt{n}}{\sqrt{m}}$$

$\square$

*Proof.* (of lemma 2.3) We have that $\Pr(|X_i - \mu| > k\sigma) \le \frac{1}{k^2}$. It follows that the probability that $\ge \frac{n}{2}$ of $X_1,\ldots,X_n$ fall outside of the segment $(\mu - k\sigma, \mu + k\sigma)$ is bounded by

$$\binom{n}{\lceil n/2 \rceil}\left(\frac{1}{k^2}\right)^{\lceil n/2 \rceil} < 2^n\left(\frac{1}{k^2}\right)^{\lceil n/2\rceil} \le \left(\frac{2}{k}\right)^n$$

$\square$

*Proof.* (of lemma 3.4) Fix a set $A \subset \mathcal{X}$. Let $(\mathcal{C}, \Omega, \mu)$ be a $\left(n\lceil\epsilon^{-2}\rceil, \epsilon\right)$-compressor for $\mathcal{H}$. Let $\tilde{\mathcal{H}}$ be the range of $\mathcal{C}$. Note that $\left|\tilde{\mathcal{H}}\right| \le 2^{n\lceil\epsilon^{-2}\rceil}$. Fix $h \in \mathcal{H}$. It is enough to show that there is $\tilde{h} \in \tilde{\mathcal{H}}$ with $\mathbb{E}_{x\in A}\left(h(x) - \tilde{h}(x)\right)^2 \le \epsilon^2$. Indeed,

$$\mathop{\mathbb{E}}_{\omega\sim\mu}\mathop{\mathbb{E}}_{x\in A}(h(x) - (\mathcal{C}_\omega h)(x))^2 = \mathop{\mathbb{E}}_{x\in A}\mathop{\mathbb{E}}_{\omega\sim\mu}(h(x) - (\mathcal{C}_\omega h)(x))^2 \le \epsilon^2.$$

Hence, there exists $\tilde{h} \in \tilde{\mathcal{H}}$ for which $\mathbb{E}_{x\in A}\left(h(x) - \tilde{h}(x)\right)^2 \le \epsilon^2$

$\square$

*Proof.* (of lemma 3.5) Denote $k = \lceil\log(dm)\rceil$. Fix a set $A \subset \mathcal{X}$. Let $\mathcal{C}$ be a $\left(n\lceil 16\epsilon^{-2}\rceil, \frac{\epsilon}{4}\right)$-compressor for $\mathcal{H}$. Define

$$(\mathcal{C}'_{\omega_1,\ldots,\omega_k}h)(x) = \mathrm{med}\left((\mathcal{C}_{\omega_1}h)(x),\ldots,(\mathcal{C}_{\omega_k}f)(x)\right)$$

345   Let $\tilde{\mathcal{H}}$ be the range of $\mathcal{C}'$. Note that $\left|\tilde{\mathcal{H}}\right| \leq 2^{kn\lceil 16\epsilon^{-2}\rceil}$. Fix $h \in \mathcal{H}$. It is enough to show that there is

346   $\tilde{h} \in \tilde{\mathcal{H}}$ with $\max_{x \in A}\left\|h(x) - \tilde{h}(x)\right\|_{\infty} \leq \epsilon$. By lemma 2.3 we have that

$$\Pr_{\omega_1,\ldots,\omega_k \sim \mu}\left(\exists x \in A,\ \left|(\mathcal{C}'_{\omega_1,\ldots,\omega_k}h)(x) - h(x)\right| > \epsilon\right) < dm2^{-k} \leq 1$$

347   In particular, there exists $\tilde{h} \in \tilde{\mathcal{H}}$ for which $\max_{x \in A}\left\|h(x) - \tilde{h}(x)\right\|_{\infty} \leq \epsilon$     $\square$

348   *Proof.* (of lemma 3.8) Items 1. is straight forward. To see item 2. note that

$$
\begin{aligned}
\mathbb{E}\left(\langle \mathbf{u}, \hat{\mathbf{w}}\rangle - \langle \mathbf{u}, \mathbf{w}\rangle\right)^2 &\leq \mathbb{E}\langle \mathbf{u}, \mathbf{w}\rangle^2 \\
&= \sum_i p_i\left(\left\lceil\frac{w_i}{p_i}\right\rceil\left(\left\lfloor\frac{w_i}{p_i}\right\rfloor + 1\right)^2 + \left(1 - \left\lceil\frac{w_i}{p_i}\right\rceil\right)\left(\left\lfloor\frac{w_i}{p_i}\right\rfloor\right)^2\right)u_i^2 \\
&= \sum_i p_i\left(\left(\left\lfloor\frac{w_i}{p_i}\right\rfloor\right)^2 + 2\left\lfloor\frac{w_i}{p_i}\right\rfloor\left\lceil\frac{w_i}{p_i}\right\rceil + \left\lceil\frac{w_i}{p_i}\right\rceil\right)u_i^2 \\
&= \sum_i p_i\left(\left(\left\lfloor\frac{w_i}{p_i}\right\rfloor + \left\lceil\frac{w_i}{p_i}\right\rceil\right)^2 + \left\lceil\frac{w_i}{p_i}\right\rceil - \left\lceil\frac{w_i}{p_i}\right\rceil^2\right)u_i^2 \\
&= \sum_i p_i\left(\left(\frac{w_i}{p_i}\right)^2 + \left\lceil\frac{w_i}{p_i}\right\rceil\left(1 - \left\lceil\frac{w_i}{p_i}\right\rceil\right)\right)u_i^2 \\
&\leq \sum_i p_i\left(\left(\frac{w_i}{p_i}\right)^2 + \frac{1}{4}\right)u_i^2 \\
&\leq \frac{1}{4}\|\mathbf{u}\|_{\infty}^2 + \sum_i \frac{w_i^2 u_i^2}{p_i} \\
&\leq \frac{1}{4} + \sum_i \frac{w_i^2 u_i^2}{p_i}
\end{aligned}
$$

349   Now, since $p_i = \frac{w_i^2}{2\|\mathbf{w}\|^2} + \frac{1}{2d}$ we have

$$\sum_i \frac{w_i^2 u_i^2}{p_i} \leq \sum_i \frac{w_i^2 u_i^2}{\frac{w_i^2}{2\|\mathbf{w}\|^2}} = 2\|\mathbf{w}\|^2\sum_i u_i^2 = 2\|\mathbf{w}\|^2$$

350                                                           $\square$

351   *Proof.* (of lemma 3.9) We have $\mathbb{E}\,AX = A\,\mathbb{E}\,X = A\mathbf{x}$. Furthermore, for any $\mathbf{u} \in \mathbb{S}^{d_2-1}$,

$$\mathbb{E}\langle \mathbf{u}, AX - A\mathbf{x}\rangle^2 = \mathbb{E}\left\langle A^T\mathbf{u}, X - \mathbf{x}\right\rangle^2 \leq \|A^T\mathbf{u}\|^2\sigma^2 \leq r^2\sigma^2$$

352                                                            $\square$

*Proof.* (of lemma 3.12) 1. and 2. are straight forward. We next prove 3. By replacing each $X_i$ with $\frac{X_i}{\sigma_i}$ we can assume w.l.o.g. that $\sigma_1 = \ldots = \sigma_k = 1$. We have

$$
\begin{aligned}
\mathop{\mathbb{E}}_{X_1,\ldots,X_k} \left\langle \mathbf{u}, \prod_{i=1}^{k} X_i - \prod_{i=1}^{k} \mathbf{x}_i \right\rangle^2 &= \mathop{\mathbb{E}}_{X_1,\ldots,X_k} \left\langle \mathbf{u}, \prod_{i=1}^{k} \left( (X_i - \mathbf{x}_i) + \mathbf{x}_i \right) - \prod_{i=1}^{k} \mathbf{x}_i \right\rangle^2 \\
&= \mathop{\mathbb{E}}_{X_1,\ldots,X_k} \left\langle \mathbf{u}, \sum_{A \subset [k]} \prod_{i \in A} (X_i - \mathbf{x}_i) \prod_{i \in A^c} \mathbf{x}_i - \prod_{i=1}^{k} \mathbf{x}_i \right\rangle^2 \\
&= \mathop{\mathbb{E}}_{X_1,\ldots,X_k} \left( \left\langle \mathbf{u}, \sum_{A \subset [k]} \prod_{i \in A} (X_i - \mathbf{x}_i) \prod_{i \in A^c} \mathbf{x}_i \right\rangle - \left\langle \mathbf{u}, \prod_{i=1}^{k} \mathbf{x}_i \right\rangle \right)^2 \\
&= \mathop{\mathbb{E}}_{X_1,\ldots,X_k} \sum_{A \subset [k]} \sum_{B \subset [k]} \left\langle \mathbf{u}, \prod_{i \in A} (X_i - \mathbf{x}_i) \prod_{i \in A^c} \mathbf{x}_i \right\rangle \left\langle \mathbf{u}, \prod_{i \in B} (X_i - \mathbf{x}_i) \prod_{i \in B^c} \mathbf{x}_i \right\rangle \\
&\quad -2 \mathop{\mathbb{E}}_{X_1,\ldots,X_k} \sum_{A \subset [k]} \left\langle \mathbf{u}, \prod_{i=1}^{k} \mathbf{x}_i \right\rangle \left\langle \mathbf{u}, \prod_{i \in A} (X_i - \mathbf{x}_i) \prod_{i \in A^c} \mathbf{x}_i \right\rangle \\
&\quad + \left\langle \mathbf{u}, \prod_{i=1}^{k} \mathbf{x}_i \right\rangle^2 \\
&\stackrel{(1)}{=} \mathop{\mathbb{E}}_{X_1,\ldots,X_k} \sum_{A \subset [k]} \left\langle \mathbf{u}, \prod_{i \in A} (X_i - \mathbf{x}_i) \prod_{i \in A^c} \mathbf{x}_i \right\rangle^2 - \left\langle \mathbf{u}, \prod_{i=1}^{k} \mathbf{x}_i \right\rangle^2 \\
&\stackrel{(2)}{\leq} \sum_{A \subset [k], A \neq [k]} \left\| \mathbf{u} \prod_{i \in A^c} \mathbf{x}_i \right\|^2 \\
&= \sum_{A \subset [k], A \neq \emptyset} \left\| \mathbf{u} \prod_{i \in A} \mathbf{x}_i \right\|^2 \\
&\stackrel{(3)}{\leq} \sum_{A \subset [k], A \neq \emptyset} \prod_{i \in A} \|\mathbf{x}_i\|_\infty^2 \\
&= \prod_{i=1}^{k} \left( 1 + \|\mathbf{x}_i\|_\infty^2 \right) - \prod_{i=1}^{k} \|\mathbf{x}_i\|_\infty^2
\end{aligned}
$$

(1) If $A \neq B$, then w.l.o.g. $k \in A \setminus B$. In this case we have

$$
\begin{aligned}
\mathop{\mathbb{E}}_{X_1,\ldots,X_k} & \left\langle \mathbf{u}, \prod_{i \in A} (X_i - \mathbf{x}_i) \prod_{i \in A^c} \mathbf{x}_i \right\rangle \left\langle \mathbf{u}, \prod_{i \in B} (X_i - \mathbf{x}_i) \prod_{i \in A^c} \mathbf{x}_i \right\rangle \\
&= \mathop{\mathbb{E}}_{X_1,\ldots,X_{k-1}} \mathop{\mathbb{E}}_{X_k} \left\langle \mathbf{u}, \prod_{i \in A} (X_i - \mathbf{x}_i) \prod_{i \in A^c} \mathbf{x}_i \right\rangle \left\langle \mathbf{u}, \prod_{i \in B} (X_i - \mathbf{x}_i) \prod_{i \in B^c} \mathbf{x}_i \right\rangle \\
&= \mathop{\mathbb{E}}_{X_1,\ldots,X_{k-1}} \left\langle \mathbf{u}, \prod_{i \in B} (X_i - \mathbf{x}_i) \prod_{i \in B^c} \mathbf{x}_i \right\rangle \mathop{\mathbb{E}}_{X_k} \left\langle \mathbf{u}, \prod_{i \in A} (X_i - \mathbf{x}_i) \prod_{i \in A^c} \mathbf{x}_i \right\rangle \\
&= \mathop{\mathbb{E}}_{X_1,\ldots,X_{k-1}} \left\langle \mathbf{u}, \prod_{i \in B} (X_i - \mathbf{x}_i) \prod_{i \in B^c} \mathbf{x}_i \right\rangle \left\langle \mathbf{u}, \prod_{i \in A \setminus [k]} (X_i - \mathbf{x}_i) \overbrace{\mathop{\mathbb{E}}_{X_k} (X_k - \mathbf{x}_k)}^{=0} \prod_{i \in A^c} \mathbf{x}_i \right\rangle \\
&= 0
\end{aligned}
$$

Similarly, if $A \neq \emptyset$, then w.l.o.g. $k \in A$. In this case we have

$$
\begin{aligned}
\mathop{\mathbb{E}}_{X_1,\ldots,X_k} \left\langle \mathbf{u}, \prod_{i=1}^{k} \mathbf{x}_i \right\rangle \left\langle \mathbf{u}, \prod_{i \in A} (X_i - \mathbf{x}_i) \prod_{i \in A^c} \mathbf{x}_i \right\rangle &= \mathop{\mathbb{E}}_{X_1,\ldots,X_{k-1}} \mathop{\mathbb{E}}_{X_k} \left\langle \mathbf{u}, \prod_{i=1}^{k} \mathbf{x}_i \right\rangle \left\langle \mathbf{u}, \prod_{i \in A} (X_i - \mathbf{x}_i) \prod_{i \in A^c} \mathbf{x}_i \right\rangle \\
&= \mathop{\mathbb{E}}_{X_1,\ldots,X_{k-1}} \left\langle \mathbf{u}, \prod_{i=1}^{k} \mathbf{x}_i \right\rangle \left\langle \mathbf{u}, \prod_{i \in A \setminus [k]} (X_i - \mathbf{x}_i) \overbrace{\mathop{\mathbb{E}}_{X_k} (X_k - \mathbf{x}_k)}^{=0} \prod_{i \in A^c} \mathbf{x}_i \right\rangle
\end{aligned}
$$

(2) Fix a set $A$ that is w.l.o.g. $A = \{1, \ldots, k'\}$. We note that if $X \in \mathbb{R}^d$ is a 1-estimator to 0, then for any vector $\mathbf{z} \in \mathbb{R}^d$

$$
\mathop{\mathbb{E}}_{X} \|\mathbf{z}X\|^2 = \sum_{i=1}^{d} z_i^2 \mathop{\mathbb{E}}_{X} X_i^2 = \sum_{i=1}^{d} z_i^2 \mathop{\mathbb{E}}_{X} \langle \mathbf{e}_i, X \rangle^2 \leq \sum_{i=1}^{d} z_i^2 = \|\mathbf{z}\|^2
$$

It follows that

$$
\underset{X_1,\ldots,X_{k'-1}}{\mathbb{E}} \left\| \mathbf{u} \prod_{i=k'+1}^{k} \mathbf{x}_i \prod_{i=1}^{k'-1} (X_i - \mathbf{x}_i) \right\|^2 = \underset{X_1,\ldots,X_{k'-2}}{\mathbb{E}} \underset{X_{k'-1}}{\mathbb{E}} \left\| \mathbf{u} \prod_{i=k'+1}^{k} \mathbf{x}_i \prod_{i=1}^{k'-1} (X_i - \mathbf{x}_i) \right\|^2
$$

$$
\leq \underset{X_1,\ldots,X_{k'-2}}{\mathbb{E}} \left\| \mathbf{u} \prod_{i=k'+1}^{k} \mathbf{x}_i \prod_{i=1}^{k'-2} (X_i - \mathbf{x}_i) \right\|^2
$$

$$
\vdots
$$

$$
\leq \left\| \mathbf{u} \prod_{i=k'+1}^{k} \mathbf{x}_i \right\|^2
$$

$$
= \left\| \mathbf{u} \prod_{i \in A^c} \mathbf{x}_i \right\|^2
$$

Hence,

$$
\underset{X_1,\ldots,X_k}{\mathbb{E}} \left\langle \mathbf{u}, \prod_{i \in A} (X_i - \mathbf{x}_i) \prod_{i \in A^c} \mathbf{x}_i \right\rangle^2 = \underset{X_1,\ldots,X_{k'-1}}{\mathbb{E}} \underset{X_{k'}}{\mathbb{E}} \left\langle \mathbf{u} \prod_{i=k'+1}^{k} \mathbf{x}_i \prod_{i=1}^{k'-1} (X_i - \mathbf{x}_i), (X_{k'} - \mathbf{x}_{k'}) \right\rangle^2
$$

$$
\overset{X_{k'} \text{ is 1-estimator of } \mathbf{x}_k}{\leq} \underset{X_1,\ldots,X_{k'-1}}{\mathbb{E}} \left\| \mathbf{u} \prod_{i=k'+1}^{k} \mathbf{x}_i \prod_{i=1}^{k'-1} (X_i - \mathbf{x}_i) \right\|^2
$$

$$
\leq \left\| \mathbf{u} \prod_{i \in A^c} \mathbf{x}_i \right\|^2
$$

(3) If $\mathbf{z} = \mathbf{u} \prod_{i \in A} \mathbf{x}_i$ then for any $j \in [d]$, $|z_j| \leq |u_j| \prod_{i \in A} \|\mathbf{x}_i\|_\infty$. Hence,

$$
\|\mathbf{z}\|^2 \leq \prod_{i \in A} \|\mathbf{x}_i\|_\infty \sum_{j=1}^{d} u_j^2 = \prod_{i \in A} \|\mathbf{x}_i\|_\infty
$$

$\square$

*Proof.* (of lemma 4.2) By adding a pair of brackets around each bit, each bracketed string can be described by $2n - 1$ correctly matched pairs of brackets, and a string of length $\leq n$. As the number of ways to correctly match $k$ pairs of brackets is the Catalan number $C_k = \frac{1}{k+1}\binom{2k}{k} \leq 2^{2k}$, we have,
$|\mathcal{S}_n| \leq 2^{4n-2} 2^{n+1}$ $\square$

*Proof.* (of lemma 4.10) Fix as set $A \subset \mathcal{X}$ of size $m$. We will construct a compressor to $\mathcal{L}_{d_1,d_2,r,R} \circ \mathcal{H}$ as follows. Given $h \in \mathcal{H}$ and $W \in \mathcal{L}_{d_1,d_2,r,R}$ we first pay a seed cost $n_s(m)$ to use $\mathcal{H}$'s compressor. Then, we use $\mathcal{H}$'s compressor to generate a $\sqrt{\frac{1}{k_1}}$-estimator $\hat{h}$ of $h$, at the cost of $k_1 n(m)$ bits. Then, we take $\hat{W}$ to be a $k_2$-sketch of $W$, at the costs of $k_2 O\left(\log\left(d_1 d_2 R + 1\right)\right)$ bits. Finally, we output the estimator $\hat{h} \circ \hat{W}$. Fix $a \in A$. We must show that $\hat{W}X := \hat{W}\hat{h}(a)$ is a 1-estimator of $\mathbf{x} = h(a)$.

Indeed, for $\mathbf{u} \in \mathbb{S}^{d_2-1}$ we have,

$$
\underset{X}{\mathbb{E}}\,\underset{\hat{W}}{\mathbb{E}}\left\langle \mathbf{u}, \hat{W}X - W\mathbf{x}\right\rangle^2 = \underset{X}{\mathbb{E}}\,\underset{\hat{W}}{\mathbb{E}}\left\langle \mathbf{u}, \hat{W}X - WX\right\rangle^2 + 2\left\langle \mathbf{u}, \hat{W}X - WX\right\rangle\left\langle \mathbf{u}, WX - W\mathbf{x}\right\rangle + \left\langle \mathbf{u}, WX - W\mathbf{x}\right\rangle^2
$$

$$
= \underset{X}{\mathbb{E}}\,\underset{\hat{W}}{\mathbb{E}}\left\langle \mathbf{u}, \hat{W}X - WX\right\rangle^2 + 2\underset{X}{\mathbb{E}}\left\langle \mathbf{u}, \overbrace{\underset{\hat{W}}{\mathbb{E}}\left[\hat{W}-W\right]}^{=0}X\right\rangle\left\langle \mathbf{u}, WX - W\mathbf{x}\right\rangle + \underset{X}{\mathbb{E}}\,\underset{\hat{W}}{\mathbb{E}}\left\langle \mathbf{u}, WX - W\mathbf{x}\right\rangle^2
$$

$$
= \underset{X}{\mathbb{E}}\,\underset{\hat{W}}{\mathbb{E}}\left\langle \mathbf{u}, \hat{W}X - WX\right\rangle^2 + \left\langle \mathbf{u}, WX - W\mathbf{x}\right\rangle^2
$$

$$
= \underset{X}{\mathbb{E}}\,\underset{\hat{W}}{\mathbb{E}}\left\langle \hat{W} - W, X\mathbf{u}^T\right\rangle^2 + \left\langle W^T\mathbf{u}, X - \mathbf{x}\right\rangle^2
$$

$$
\overset{\text{Lemma 3.8}}{\leq} \frac{2\|W\|_F^2 + 1}{k_2}\underset{X}{\mathbb{E}}\|X\|^2 + \frac{1}{k_1}\|W\mathbf{u}\|^2
$$

$$
\overset{(1)}{\leq} \frac{2\|W\|_F^2 + 1}{k_2}\left[\underset{X}{\mathbb{E}}\|X-\mathbf{x}\|^2 + \|\mathbf{x}\|^2\right] + \frac{1}{k_1}\|W\|^2
$$

$$
\overset{(2)}{\leq} \frac{2\|W\|_F^2 + 1}{k_2}\left[\frac{1}{k_1}d_1 + \|\mathbf{x}\|^2\right] + \frac{1}{k_1}\|W\|^2
$$

$$
\leq \frac{2R^2 + 1}{k_2}\left[\frac{1}{k_1}d_1 + B^2\right] + \frac{1}{k_1}r^2
$$

(1) We have

$$
\underset{X}{\mathbb{E}}\|X-\mathbf{x}\|^2 = \underset{X}{\mathbb{E}}\|X\|^2 - 2\langle X, \mathbf{x}\rangle + \|\mathbf{x}\|^2
$$

$$
= \underset{X}{\mathbb{E}}\|X\|^2 - 2\langle \mathbb{E}\,X, \mathbf{x}\rangle + \|\mathbf{x}\|^2
$$

$$
= \underset{X}{\mathbb{E}}\|X\|^2 - \|\mathbf{x}\|^2
$$

(2) We have

$$
\underset{X}{\mathbb{E}}\|X-\mathbf{x}\|^2 = \sum_{i=1}^{d_1}\mathbb{E}(X_i - x_i)^2
$$

$$
= \sum_{i=1}^{d_1}\mathbb{E}\left\langle X-\mathbf{x}, \mathbf{e}_i\right\rangle^2
$$

$$
\leq \sum_{i=1}^{d_1}\frac{1}{k_1} = \frac{d_1}{k_1}
$$

Finally, by choosing $k_1 = \lceil 2r^2\rceil + 1$ and $k_2 = 2(d_1 + B^2)(2R^2 + 1)$ we get the result. $\qquad\square$

**Lemma 6.2.** *Suppose that $\{X_n\}_{n=1}^{\infty}$ are independent $\sigma$-estimators to $\mathbf{x} \in \mathbb{R}^d$. Let $\rho(\mathbf{t}) = \sum_{n=0}^{\infty}\mathbf{a}_n\mathbf{t}^n$. Let $U = \mathbf{a}_0 + \sum_{n=1}^{\infty}\hat{\mathbf{a}}_n Y_n$ where $Y_n = \prod_{i=1}^{n}X_i$ and $\hat{\mathbf{a}}_n = \frac{\mathbf{a}_n}{p_1}$ w.p. $p_i$ and $0$ otherwise.*

*Then $U$ is $\sigma'$-estimator of $\rho(\mathbf{x})$ with $\sigma' = \sum_{n=1}^{\infty}\sqrt{\frac{\|\mathbf{a}_n\|_\infty^2}{p_n}\left((\sigma^2 + \|\mathbf{x}\|_\infty^2)^n + (1 - p_n)d\|\mathbf{x}\|_\infty^{2n}\right)}$.*

**Remark 6.3.** *In particular, if $\|\mathbf{a}_n\|_\infty \leq B^n$, $\sqrt{\sigma^2 + \|\mathbf{x}\|_\infty^2} \leq \frac{1}{6B}$ and $p_n = \begin{cases} 1 & n \leq \left\lceil\frac{\log_3(d)}{2}\right\rceil \\ 4^{-n} & otherwise \end{cases}$,*

*We have $\sigma' \leq 1$ and $\mathbb{E}\max\{n : \hat{\mathbf{a}}_n \neq 0\} \leq \frac{\log_3(d)+4}{2}$. Indeed,*

$$
\sum_{n=1}^{\infty}\sqrt{\frac{\|\mathbf{a}_n\|_\infty^2}{p_n}\left((\sigma^2 + \|\mathbf{x}\|_\infty^2)^n + (1-p_n)d\|\mathbf{x}\|_\infty^{2n}\right)} \leq \sum_{n=1}^{\infty}\sqrt{\frac{\|\mathbf{a}_n\|_\infty^2}{p_n}(\sigma^2 + \|\mathbf{x}\|_\infty^2)^n} + \sum_{n=1}^{\infty}\sqrt{\frac{\|\mathbf{a}_n\|_\infty^2}{p_n}(1-p_n)d\|\mathbf{x}\|_\infty^{2n}}
$$

$$
\leq \sum_{n=1}^{\infty}(2B)^n\sqrt{(\sigma^2 + \|\mathbf{x}\|_\infty^2)^n} + \sqrt{d}\sum_{n=\left\lceil\frac{\log_3(d)}{2}\right\rceil+1}^{\infty}(2B)^n\|\mathbf{x}\|_\infty^n
$$

$$
\leq \sum_{n=1}^{\infty}\left(\frac{1}{3}\right)^n + \sqrt{d}\sum_{n=\left\lceil\frac{\log_3(d)}{2}\right\rceil+1}^{\infty}\left(\frac{1}{3}\right)^n
$$

$$
\leq \sum_{n=1}^{\infty}\left(\frac{1}{3}\right)^n + \sum_{n=1}^{\infty}\left(\frac{1}{3}\right)^n = 1
$$

*and*

$$\mathbb{E}\max\{n : \hat{\mathbf{a}}_n \neq 0\} \leq \left\lceil \frac{\log_3(d)}{2} \right\rceil + \sum_{n=\left\lceil \frac{\log_3(d)}{2} \right\rceil + 1}^{\infty} 4^{-n} n \leq \left\lceil \frac{\log_3(d)}{2} \right\rceil + 1$$

*Proof.* By lemma 3.12 it is enough to show that for all $n$, $\hat{\mathbf{a}}_n Y_n$ is a $\sqrt{\frac{\|\mathbf{a}_n\|_\infty^2}{p_n}\left((\sigma^2 + \|\mathbf{x}\|_\infty^2)^n + (1-p_n)d\|\mathbf{x}\|_\infty^{2n}\right)}$-estimator of $\mathbf{a}_n \mathbf{x}^n$. Indeed,

$$
\begin{aligned}
\mathrm{VAR}\left(\langle \mathbf{u}, \hat{\mathbf{a}}_n Y_n \rangle\right) &= \mathbb{E}\left(\langle \mathbf{u}, \hat{\mathbf{a}}_n Y_n \rangle - \langle \mathbf{u}, \mathbf{a}_n \mathbf{x}^n \rangle\right)^2 \\
&= p_n \mathbb{E}\left(\left\langle \mathbf{u}, \frac{\mathbf{a}_n}{p_n} Y_n \right\rangle - \langle \mathbf{u}, \mathbf{a}_n \mathbf{x}^n \rangle\right)^2 + (1-p_n)\langle \mathbf{u}, \mathbf{a}_n \mathbf{x}^n \rangle^2 \\
&= \frac{1}{p_n}\mathbb{E}\langle \mathbf{u}, \mathbf{a}_n Y_n \rangle^2 - 2\mathbb{E}\langle \mathbf{u}, \mathbf{a}_n Y_n \rangle\langle \mathbf{u}, \mathbf{a}_n \mathbf{x}^n \rangle + p_n \langle \mathbf{u}, \mathbf{a}_n \mathbf{x}^n \rangle^2 + (1-p_n)\langle \mathbf{u}, \mathbf{a}_n \mathbf{x}^n \rangle^2 \\
&= \frac{1}{p_n}\mathbb{E}\langle \mathbf{u}, \mathbf{a}_n Y_n \rangle^2 - \langle \mathbf{u}, \mathbf{a}_n \mathbf{x}^n \rangle^2 \\
&= \frac{1}{p_n}\mathbb{E}\left(\langle \mathbf{a}_n \mathbf{u}, Y_n \rangle^2 - \langle \mathbf{a}_n \mathbf{u}, \mathbf{x}^n \rangle^2\right) + \frac{1-p_n}{p_n}\langle \mathbf{u}, \mathbf{a}_n \mathbf{x}^n \rangle^2 \\
&\overset{\text{lemma 3.12}}{\leq} \frac{\|\mathbf{a}_n \mathbf{u}\|_2^2}{p_n}\left(\left(\sigma^2 + \|\mathbf{x}\|_\infty^2\right)^n + (1-p_n)\|\mathbf{x}^n\|_2^2\right) \\
&\leq \frac{\|\mathbf{a}_n\|_\infty^2}{p_n}\left(\left(\sigma^2 + \|\mathbf{x}\|_\infty^2\right)^n + (1-p_n)d\|\mathbf{x}\|_\infty^{2n}\right)
\end{aligned}
$$

$\square$

*Proof.* (of lemma 4.13) Consider the complex function $f(z) = \frac{e^z}{1+e^z}$. It is defined in the strip $\{z = x + iy : |y| < \pi\}$. By Cauchy integral formula, for any $r < \pi$, $a \in \mathbb{R}$ and $n \geq 0$,

$$f^{(n)}(a) = \frac{n!}{2\pi i}\int_{|z-a|=r}\frac{f(z)}{(z-a)^{n+1}}$$

It follows that

$$\left|f^{(n)}(a)\right| \leq \frac{n!}{r^n}\max_{|z-a|=r}|f(z)| \leq \frac{n!}{r^n}\max_{x+iy:|y|<r}|f(x+iy)|$$

Now, if $|y| < r < \frac{\pi}{2}$, we have

$$|f(x+iy)| = \frac{e^x}{|1+e^{iy}e^x|} \leq \frac{e^x}{|1+\cos(y)e^x|} \leq \frac{e^x}{|1+\cos(r)e^x|} \leq \frac{1}{\cos(r)}$$

This implies that $\frac{e^x}{1+e^x}$ is strongly bounded. Likewise, since $\frac{e^x}{1+e^x}$ is the derivative of $\ln(1+e^x)$, the function $\ln(1+e^x)$ is strongly bounded as well. $\square$

*Proof.* (of lemma 4.14) Fix a set $A \subset \mathcal{X}$ of size $\leq m$. Let $\epsilon^2 = \sigma^2 = \frac{1}{72B^2}$ and note that $\sqrt{\sigma^2 + \epsilon^2} \leq \frac{1}{6B}$. To generate a 1-estimator to $\rho \circ h \in \rho \circ \mathcal{H}$ on $A$ we first describe $\tilde{h}$, which forms the seed, such that $\forall i \in [m]$, $\|\tilde{h}(x_i) - h(x_i)\|_\infty \leq \epsilon$. Then, we generate $\sigma$-estimators $\hat{h}_1, \hat{h}_2, \ldots,$ to $h|_A$. Finally, we sample Bernoulli random variables $Z_1, Z_2, \ldots$ where the parameter of $Z_n$ is

$$p_n = \begin{cases} 1 & n \leq \left\lceil \frac{\log_3(d)}{2} \right\rceil \\ 4^{-n} & \text{otherwise} \end{cases}. \text{ The final estimator is}$$

$$\hat{g}(x) = \rho(\tilde{h}(x)) + \sum_{n=1}^{\infty}\frac{\rho^{(n)}(\tilde{h}(x))}{n!}\frac{Z_n}{p_n}Y_n \text{ where } Y_n = \prod_{i=1}^{n}\left(\hat{h}_i(x) - \tilde{h}(x)\right)$$

By lemma 6.2 and the following remark, $\hat{g}$ is 1-estimator of $\rho \circ h|_A$.

How many bits do we need in order to specify $\hat{g}$? By lemma 4.6 the restriction of $\mathcal{H}|_A$ has an $\epsilon$-cover, w.r.t. the $\infty$-norm, of log-size $\lesssim n_s(m) + \frac{n(m)\log(md)}{\epsilon^2}$. So the generation of the seed $\tilde{h}$ costs $n_s(m) + \frac{n(m)\log(md)}{\epsilon^2}$ bits. We also need to specify $N := \max\{n : Z_n \neq 0\}$, $Z_1, \ldots, Z_N$ and $\hat{h}_1, \ldots, \hat{h}_N$. This can be done by concatenating the descriptions of the pairs $(Z_n, \hat{h}_n)$ for $n = 1, \ldots, N$. The bit cost of this is bounded (in expectation) by $\frac{\log_3(d)+4}{2}\left(\lceil 72B^2 \rceil n(m) + 1\right)$ $\square$