[Reviews · NeurIPS 2019]

Reviewer 1



The major aim of this paper is to provide improved bounds for the sample complexity of the networks. To this end, the authors resort to a technique that takes into consideration the magnitudes of the weights. Even if similar techniques have been used previously in machine learning, the way in which the method is applied in here seems to be new. The novelty stems from the use of the approximate description length in proving the theoretical results. The authors provide a fair interpretation of the obtained results. It is hard to anticipate the impact of these results on the practitioners who are employing networks in their works. It is likely that the results presented in this manuscript to be of interest for other researchers who are focused on theoretical aspects related to neural networks. ******************************************* I have decided for this submission to keep the same score as in the first phase of the review process.

Reviewer 2



In this paper the authors establish upper bounds on the generalization error of classes of norm-bounded neural networks. There is a long line of literature on this exact question, and this paper claims to resolve an interesting open question in this area (at least when the depth of the network is viewed as a constant). In particular, the paper considers generalization bounds for a class of fully-connected networks of constant depth and whose matrices are of bounded norm. Work by Bartlett et al. ("Spectrally normalized margin bounds on neural networks", ref [4] in the paper) proved an upper bound on generalization error that contains a factor growing as the (1,2)-matrix norm of any layer. If one further assumes that the depth as well as all the spectral norms are constants, then this is the dominant term (up to logarithmic factors) in their generalization bound. In this regime, the main result of this paper improves this (1,2)-matrix norm to the Frobenius norm (i.e., (2,2)-matrix norm). The authors achieve this by defining a notion of approximate description length of a class of functions, which roughly speaking, describes how many bits one needs to compress a function in this class to in order to be able to approximately recover the network from these bits. Overall the paper is rather clearly written. One concern I have is that there is not a lot of discussion (e.g., in the introduction) about what in their technique allows them to improve upon the previous work of Bartlett et al, or in their language, to remove the additional factor of d (depth), from sample complexity bounds. This would give more confidence in their result. In particular, something describing the following might be useful: their proof proceeds in a similar pattern to previous covering number-based upper bounds on generalization error of neural networks, except by replacing utilization of covering numbers with approximate description length. The key in their proof that allows them to improve the bound of Bartlett at al. seems to be Lemma 3.6, which allows one to replace the (2,1)-norm in covering number bounds for linear maps with the Frobenius norm (and using approximate description length instead of covering number). At least part of the reason for that improvement seems to result from the fact that they measure the covering number in the target with respect to the L_infty as opposed to L_2 norm (Defn 2.1). If this is the case, I wonder if the proof of the main result can be made to only use the language of covering numbers? Finally, there is some work appearing after the Bartlett et al. paper [4] establishing generalization bounds for networks with various bounds on norms and parameters (e.g., [LLW+18] below). Some discussion of these papers might be useful. [LLW+18] Xingguo Li, Junwei Lu, Zhaoran Wang, Jarvis Haupt, and Tuo Zhao. On tighter generalization bound for deep neural networks: CNNs, Resnets, and beyond. arXiv:1806.05159, 2018. I did not verify correctness of proofs in the supplementary material. Overall, this feels like a somewhat significant contribution that introduces some new ideas to resolve an open question. Miscellaneous comments: In the equation of Theorem 3.5, I think the middle term should be dropped (or flipped with the first term). (L_infty covering numbers are always at least the L_2 covering numbers.) Line 149: samples -> sampled Line 159: that that -> that Line 170, 2nd word: to -> the Line 186: I think the sigma at the beginning of the line should be rho? Update: I was already quite positive about the paper, and nothing in the authors' response changed my opinion.

Reviewer 3



Although the proposed analysis is quite elegant, a natural question is whether it has to be this sophisticated. A large portion of the efforts was devoted to bounding the description length of compositing an arbitrary function with the activation function \rho. I noticed that all activation functions studied in the paper are contraction functions. That is, for any t and t', |\rho(t) - \rho(t')| <= |t - t'|. I am wondering if the proof can be greatly simplified (or the description length argument can be completely circumstanced) if this property is properly used. In particular, if matrix W' is an approximation of W such that ||W' - W|| <= \eps1, and two vectors x' and x satisfies ||x' - x|| <= \eps2, then ||\rho(W'x') - \rho(Wx)|| <= ||W'x' - Wx|| <= ||W' - W||*||x'|| + ||W||*||x - x'|| <= \eps1*||x'|| \eps2 * ||W||. It means that we can upper bound the difference of the output (\rho(W'x') - \rho(Wx)) using an upper bound on the input (||x' - x||). By recursively applying this inequality layer by layer, it leads to a function approximation bound, and can be used to derive a covering number bound and finally a generalization bound. I got a feeling that this bound is also sub-linear to the number of parameters, but I am not sure. Hopefully the authors can convince me by the rebuttal that I am wrong. == I have read the authors' rebuttal. I am not totally convinced but I tend to believe that it's true. My score remains unchanged.

[Author Response · NeurIPS 2019]

We thank the reviewers for their positive reviews. We will surely take their comments into account , and amend the paper accordingly.

As for the question of reviewer # 3, the problem with the suggested approach is that in order to have a suitable $\epsilon$-cover of the space of d times d matrices, we need $e^{\Omega\left((d/\epsilon)^2\right)}$ points. I think that the best way to be convinced that this approach would not work, is that this is the approach of virtually all previous papers.

[Meta-Review · NeurIPS 2019]

This paper proposes a new framework for bounding the generalization error of fully connected neural nets. The authors are able to show that, for sufficiently smooth activation functions, the number of examples required to achieve a good generalization error scales sublinearly with the total number of parameters in the network. This is a significantly better bound than the previous state-of-the-art results. The analytical tools based on description length are very interesting, and could be applicable to the analysis of other multi-layer non-convex models. All three reviewers are uniformly enthusiastic about this work, which is guaranteed to attract a great deal of attention and to catalyze further research activity.